# Discovery of Candidate Stool Biomarker Proteins for Biliary Atresia Using Proteome Analysis by Data-Independent Acquisition Mass Spectrometry

**DOI:** 10.3390/proteomes8040036

**Published:** 2020-11-27

**Authors:** Eiichiro Watanabe, Yusuke Kawashima, Wataru Suda, Tomo Kakihara, Shinya Takazawa, Daisuke Nakajima, Ren Nakamura, Akira Nishi, Kan Suzuki, Osamu Ohara, Jun Fujishiro

**Affiliations:** 1Department of Pediatric Surgery, Faculty of Medicine, The University of Tokyo, Tokyo 113-8655, Japan; watanabee-psu@h.u-tokyo.ac.jp (E.W.); tomo.kakihara@riken.jp (T.K.); SUZUKIK-PSU@h.u-tokyo.ac.jp (K.S.); 2Department of Applied Genomics, Kazusa DNA Research Institute, Kisarazu 292-0818, Japan; ykawashi@kazusa.or.jp (Y.K.); nakajima@kazusa.or.jp (D.N.); r-nakamu@kazusa.or.jp (R.N.); ohara@kazusa.or.jp (O.O.); 3Laboratory for Microbiome Sciences, RIKEN Center for Integrative Medical Sciences, Yokohama 230-0045, Japan; wataru.suda@riken.jp; 4Department of Surgery, Gunma Children’s Medical Center, Shibukawa 277-8577, Japan; shinya.takazawa@gmail.com (S.T.); anishi@gcmc.pref.gunma.jp (A.N.)

**Keywords:** biliary atresia, proteome analysis, DIA–MS, stool biomarker

## Abstract

Biliary atresia (BA) is a destructive inflammatory obliterative cholangiopathy of the neonate that affects various parts of the bile duct. If early diagnosis followed by Kasai portoenterostomy is not performed, progressive liver cirrhosis frequently leads to liver transplantation in the early stage of life. Therefore, prompt diagnosis is necessary for the rescue of BA patients. However, the prompt diagnosis of BA remains challenging because specific and reliable biomarkers for BA are currently unavailable. In this study, we discovered potential biomarkers for BA using deep proteome analysis by data-independent acquisition mass spectrometry (DIA–MS). Four patients with BA and three patients with neonatal cholestasis of other etiologies (non-BA) were recruited for stool proteome analysis. Among the 2110 host-derived proteins detected in their stools, 49 proteins were significantly higher in patients with BA and 54 proteins were significantly lower. These varying stool protein levels in infants with BA can provide potential biomarkers for BA. As demonstrated in this study, the deep proteome analysis of stools has great potential not only in detecting new stool biomarkers for BA but also in elucidating the pathophysiology of BA and other pediatric diseases, especially in the field of pediatric gastroenterology.

## 1. Introduction

Biliary atresia (BA) is a destructive inflammatory obliterative cholangiopathy of neonates that affects various parts of the intra- and extrahepatic bile duct and causes cholestasis, which manifests as jaundice with hyperbilirubinemia. BA is a pediatric emergency because progression frequently leads to cirrhosis and liver transplantation [1]. As there are many other rare diseases that present with cholestasis in infancy and specific biomarkers for BA have not yet been identified, the prompt diagnosis of BA remains challenging for pediatric surgeons and pediatric gastroenterologists. Operative cholangiogram, an invasive diagnostic procedure, is the gold standard for the definitive diagnosis of BA, as it distinguishes BA from other causes of neonatal cholestasis (non-BA) [1]. Therefore, a simple and noninvasive examination with sufficient accuracy and reliability using biomarkers for BA is urgently required.

Plasma and serum are used as samples for biomarker discovery because they can be collected with low invasiveness. However, plasma/serum collection sometimes becomes a burden, especially in small children. On the other hand, stool is an ideal sample for discovering new biomarkers for pediatric diseases because it can be obtained noninvasively from patients. As an analysis that uses stool samples, the Sudan III stain of stool fat is available for the early diagnosis of BA, but its specificity is not sufficient [2]. In addition, gut microbiota analysis by 16S rRNA sequencing has been attracting attention in recent years and has been used for biomarker discovery in various diseases. The relevant microbiota, which is a potential diagnostic marker for BA, has also been reported [3,4]. On the other hand, the clinical application of these biomarkers has problems, such as large individual differences in gut microbiota [5] and high costs of the analysis. Thus, we focused on host proteins rather than bacteria in stools. Stools are formed via the gastrointestinal tract, which contains many host proteins that originate from gastrointestinal tissues [6]. Therefore, stool is a feasible clinical sample for exploring biomarkers of gastrointestinal diseases such as BA. This expectation also originated from one of the clinical symptoms of cholestasis, such as BA, in which the stool color gradually changed from normal (yellowish, brown, and greenish) to abnormal (clay-colored, pale yellowish, and light yellowish), followed by a gradual reduction in bile supply from the liver and gallbladder [7]. In actuality, host proteins in the stool of patients with colorectal cancer have been examined using proteomics, leading to the discovery of candidate colon biomarkers [8,9,10]. For proteomic analysis of stools, optimized methods have not been established, and up to one thousand host proteins are detectable in stools [6]. Therefore, high-depth proteome analysis is required to detect more host-derived proteins in stools.

Regarding proteomics technologies, data-dependent acquisition mass spectrometry (DDA–MS)-based proteomics is the mainstream; however, in recent years, data-independent acquisition mass spectrometry (DIA–MS)-based proteomics has attracted attention. Single-shot analysis by DIA–MS enables deep proteome analysis with higher reproducibility than DDA–MS and is suitable for biomarker discovery [11,12,13]. In particular, overlapping-window DIA–MS reduces the complexity of the MS2 spectrum, allowing deeper proteome analysis than normal DIA–MS [14,15]. In addition, the Prosit web tool enables the creation of high-quality spectral libraries for the identification of MS2 spectra in DIA–MS from protein sequence databases, which eliminates the need to create spectral libraries by DDA–MS [16]. In this way, the DIA–MS-based proteome analysis technologies have been refined, and it has become possible to easily perform deep proteome analysis.

In this study, we focused on host-derived proteins in stools and established an overlapping window DIA-based proteome analysis using noninvasive stool samples for the discovery of new stool biomarkers for BA. We successfully detected over 2000 host-derived proteins in stool samples. This is the first research article that attempts to discover new biomarkers for BA using deep proteome analysis of stools.

## 2. Methods

### 2.1. Patients

This was a retrospective observational study that analyzed stool proteins of BA and non-BA patients. Four BA patients before Kasai portoenterostomy and three non-BA patients were recruited for stool proteome analysis. Naturally defecated stools were preserved at −80 °C. All patients with BA were classified as type III (obstruction of the most proximal part of the extrahepatic biliary tract at the porta hepatis), whereas three non-BA patients had neonatal intrahepatic cholestasis caused by citrin deficiency (NICCD), cholestasis after repair of gastroschisis, and veno-occlusive disease (VOD). The average ages of BA patients and non-BA patients at the time of stool collection were 56 and 80 days, respectively (Table 1).

### 2.2. Proteome Analysis

Soluble proteins in stools prepared in PBS with protease inhibitors were extracted by pipetting and inverting after incubating for 30 min on ice. After centrifugation at 15,000× *g* for 15 min at 4 °C to remove insoluble matter, the supernatants were transferred to new tubes and subjected to trichloroacetic acid precipitation (final concentration 12.5% *v/v*), followed by acetone washing and drying via an opened lid. The dried samples were redissolved in 0.5% sodium dodecanoate and 100 mM Tris-HCl (pH 8.5) using a water-bath-type sonicator (Bioruptor UCD-200, SonicBio Corporation, Kanagawa, Japan). The pretreatment for shotgun proteome analysis was performed as reported previously [15].

Peptides were directly injected onto a 75 μm × 20 cm PicoFrit emitter (New Objective, Woburn, MA, USA) packed in-house with C18 core-shell particles (CAPCELL CORE MP 2.7 μm, 160 Å material; Osaka Soda Co., Ltd., Osaka, Japan) at 45 °C and then separated with an 80-min gradient at a flow rate of 100 nL/min using an UltiMate 3000 RSLCnano LC system (Thermo Fisher Scientific, Waltham, MA, USA). Peptides that eluted from the column were analyzed on a Q Exactive HF-X (Thermo Fisher Scientific) for overlapping window DIA–MS [15]. MS1 spectra were collected in the range of 495–785 *m/z* at 30,000 resolutions to set an automatic gain control target of 3e6 and maximum injection time of 55. MS2 spectra were collected in the range of more than 200 *m/z* at 30,000 resolutions to set an automatic gain control target of 3e6, maximum injection time of “auto”, and stepped normalized collision energies of 22, 26, and 30%. The isolation width for MS2 was set to 4 *m/z*, and overlapping window patterns in 500–780 *m/z* using window placements were optimized using Skyline.

MS files were searched for a human spectral library using Scaffold DIA (Proteome Software, Inc., Portland, OR, USA). The human spectral library was generated from the human protein sequence database (UniProt id UP000005640, reviewed, canonical) using Prosit [16]. The Scaffold DIA search parameters were as follows: experimental data search enzyme, trypsin; maximum missed cleavage sites, 1; precursor mass tolerance, 8 ppm; fragment mass tolerance, 8 ppm; and static modification, cysteine carbamidomethylation. The protein identification threshold was set for both peptide and protein false discovery rates of less than 1%. Peptide quantification was calculated using the EncyclopeDIA algorithm [17] in Scaffold DIA. For each peptide, the four highest-quality fragment ions were selected for quantitation. Protein quantification was estimated from the summed peptide quantification.

### 2.3. Data Analysis

The Gene Ontology (GO) enrichment analysis tool (Enrichr) was used to retrieve functional annotation [18]. To determine differential proteins between BA and non-BA, the statistical *p*-value (Mann–Whitney U test, *p* < 0.05) was used in data analysis. Elevated proteins in plasma and liver tissues were referred to the Human Protein Atlas (HPA; https://www.proteinatlas.org/) [19]. A heatmap was drawn based on Z-scores calculated from the DIA protein quantification using the R (version 3.5.1) function “heatmap2”.

### 2.4. Ethical Approval and Consent to Participate

This study was approved by the institutional review board (IRB) of the Faculty of Medicine, University of Tokyo (IRB No. 2019010NI), and informed consent was obtained from all subjects.

## 3. Results and Discussion

Figure 1 shows the stool proteome analysis workflow. In this study, to prevent entry of proteins other than those derived from the host as much as possible, proteins were mildly extracted with PBS so as not to break the bacteria and food debris in the stool, and proteins were purified by TCA precipitation. Then, the proteins were digested and measured using overlapping DIA–MS. Stool proteins were identified and quantified from the seven MS data (four samples from patients with BA and three from non-BA individuals). In our proteome analysis, 2110 host-derived proteins were identified in stool samples. The host stool proteins overlapped only approximately 50% with the plasma proteins, and the plasma and stools had different protein profiles (Figure 2A). Thus, unique stool biomarkers may be identified. In addition, a wide range of proteome analysis was performed with dynamic ranges of 10^7^ or greater, and the number of identified proteins was approximately 2000 (Figure 2B). When measuring HEK293 digests in the same analysis, more than 8000 proteins were observed. Based on these facts, the host stool protein concentrations presented a wide dynamic range, and high-depth analyses such as DIA–MS have great value for discovering biomarkers from stool samples.

Among the identified proteins, 103 were significantly different (*p* < 0.05) between the two groups (BA vs. non-BA) (Figure 3). Of these 103 proteins, 49 proteins were significantly higher in patients with BA (BA-dominant: Table 2), whereas 54 proteins were significantly lower in patients with BA (non-BA-dominant: Table 3).

GO enrichment analysis demonstrated that palmitoyl-CoA hydrolase activity (GO:0016290), exo-alpha-sialidase activity (GO:0004308), alpha-sialidase activity (GO:0016997), beta-galactosidase activity (GO:0004565), and palmitoyl-(protein) hydrolase activity (GO:0008474) were the top five combined molecular function scores in the BA-dominant stools (Figure 4A). Furthermore, carcinoembryonic antigens such as carcinoembryonic antigen-related cell adhesion molecule (CEACAM) 1, CEACAM5, and CEACAM8 were included in the 49 dominant proteins in the stools of the BA group. In addition to CEACAM1, proteins that were elevated in liver tissues, such as chitinase-3-like protein 1(CHI3L1), xanthine dehydrogenase/oxidase (XDH), C4b-binding protein alpha chain (C4BPA), microsomal triglyceride transfer protein large subunit (MTTP), estradiol 17-beta-dehydrogenase 2 (HSD17B2), apical endosomal glycoprotein (MAMDC4), alkaline phosphatase, tissue-nonspecific isozyme (ALPL), and sigma non-opioid intracellular receptor 1(SIGMAR1) were also included in the 49 dominant proteins in the stools of the BA group [19].

GO enrichment analysis also revealed that carbohydrate kinase activity (GO:0019200), aldehyde dehydrogenase (NAD) activity (GO:0004029), retinal dehydrogenase activity (GO:0001758), 3-chloroallyl aldehyde dehydrogenase activity (GO:0004028), and oxidoreductase activity, acting on a sulfur group of donors, with disulfide as acceptor (GO:0016671), were the top five combined molecular function scores in the non-BA-dominant stools (Figure 4B). In addition, proteins that were elevated in liver tissues, such as retinol-binding protein 4 (RBP4), serine hydroxymethyltransferase, mitochondrial (SHMT2), hydroxymethylglutaryl-CoA synthase, cytoplasmic (HMGCS1), alcohol dehydrogenase 6 (ADH6), retinal dehydrogenase 1 (ALDH1A1), short-chain specific acyl-CoA dehydrogenase, mitochondrial (ACADS), adenosine kinase (ADK), ketohexokinase (KHK), 3-ketoacyl-CoA thiolase, mitochondrial (ACAA2), phosphoserine aminotransferase (PSAT1), alpha-methylacyl-CoA racemase (AMACR), and prostaglandin reductase 1 (PTGR1) were included in the 54 dominant proteins in the stools of the non-BA group [19].

BA is a disorder that occurs during infancy with unknown etiology, which may lead to liver cirrhosis [1]. BA requires prompt and accurate diagnosis because late Kasai portoenterostomy is one of the risk factors of inappropriate bile drainage, which is an early indication for liver transplantation [20]. However, neonatal cholestasis has many causes (other than BA); thus, the accurate diagnosis of BA is challenging. Although many examinations are available for diagnosing BA, including Sudan III staining of stool fat [2], measurement of duodenal bile acid [21], and hepatobiliary scintigraphy [22], more reliable examinations or biomarkers for BA are needed. Unfortunately, invasive procedures, such as operative cholangiogram [1], are eventually required to distinguish BA from non-BA causes of neonatal cholestasis, as in non-BA.1 patient in this study, who underwent an operative cholangiogram in order to distinguish the condition from BA.

Many studies have attempted to determine the etiology of BA and discover new specific BA biomarkers. Recently, a serum proteome analysis of patients with BA found that serum levels of matrix metalloprotease-7 (MMP-7) are high in patients with BA, and this has been considered a feasible BA biomarker [23,24,25]. These studies suggested that serum MMP-7 may help diagnose BA, but the diagnostic range of enzyme-linked immunosorbent assay (ELISA) kits for MMP-7 is inconsistent. Therefore, the role of MMP-7 as a feasible biomarker remains controversial [26].

We hypothesized that stools of patients with BA contained fewer proteins produced in the biliary tract, secondary to the obstruction to the normal route of the bile juice, which is the pathophysiological mechanism of BA. Therefore, certain specific proteins that originate in the biliary tract are possibly absent or dramatically reduced in the stools of patients with BA, compared to the stools of non-BA patients.

We found that specific proteins that are elevated in the liver tissues, such as RBP4, SHMT2, HMGCS1, ADH6, ALDH1A1, ACADS, ADK, KHK, ACAA2, PSAT1, AMACR, and PTGR1, presented significantly lower abundances in stool samples from patients with BA. This finding supported our hypothesis that stools in patients with BA contained fewer proteins produced in the biliary tract. Measurement of these specific proteins or a combination of these proteins may assist in the early diagnosis of BA.

In addition, among the 49 dominant proteins in the stools of the BA group, we found that specific proteins such as CEACAM1, CEACAM5, and CEACAM8 were significantly higher in patients with BA than in non-BA patients. CEACAM1, known as biliary glycoprotein (BGP-I) is considered a cell adhesion molecule that is also distributed in the biliary tract [19,27,28]. Furthermore, soluble CEACAM1 is shed into human bile, where it can serve as an indicator of obstructive and inflammatory liver diseases [28]. Regarding CEACAM5, components of human bile from patients with biliary obstruction exhibited cross-reactivity with CEACAM5 antisera in the absence of gastrointestinal malignancies [29]. Moreover, CHI3L1 is correlated with worse outcomes of BA [30], and XDH plays a role in oxidative stress and hepatic disease pathogenesis [31]. Therefore, these dominant proteins in the stools of the BA group may contribute to the pathogenesis of BA, and measurement of these proteins in stool may also assist in the early diagnosis of BA.

This study has several limitations. First, the sample size of the preliminary study was small because BA is a rare disease, and the prevalence of BA ranges from 1 in 5000 to 25,000 live births [1]. Second, the incidence of non-BA patients is also very small. Third, although our data were statistically analyzed, a study with larger cohorts may show different results, and it is possible that these results are not generalizable to larger, more diverse populations.

However, deep proteomic analysis of BA patient stools is a new method that has the potential to detect biomarkers and elucidate the unknown etiology of BA according to stool proteins. The stool biomarker candidates of BA that were found in this study can be clinically applied after they are validated by studies that use target-based high-throughput methods, such as ELISA and selected reaction monitoring (SRM). Therefore, it is quite important to validate our preliminary results with larger cohorts.

## 4. Conclusions

Our study is the first to establish deep proteome analysis of stools and apply it to infants with cholestasis, including both BA and non-BA cohorts. Our new method of deep proteome analysis by DIA–MS can detect over 2000 host-derived proteins in stools and provides a method for discovering new BA biomarkers. Further large-scale studies are needed to validate our results regarding the varying levels of host-derived stool proteins in BA as potential specific and reliable biomarkers for diagnosing this disease. Moreover, deep proteome analysis of stools has great potential to elucidate the pathophysiology of BA and other pediatric diseases, especially in the field of pediatric gastroenterology.

Mass spectrometry proteomics data have been deposited in the ProteomeXchange Consortium (http://proteomecentral.proteomexchange.org) via the jPOST partner repository (http://jpostdb.org) with the dataset identifier PXD022391.

## Figures and Tables

**Figure 1 proteomes-08-00036-f001:**
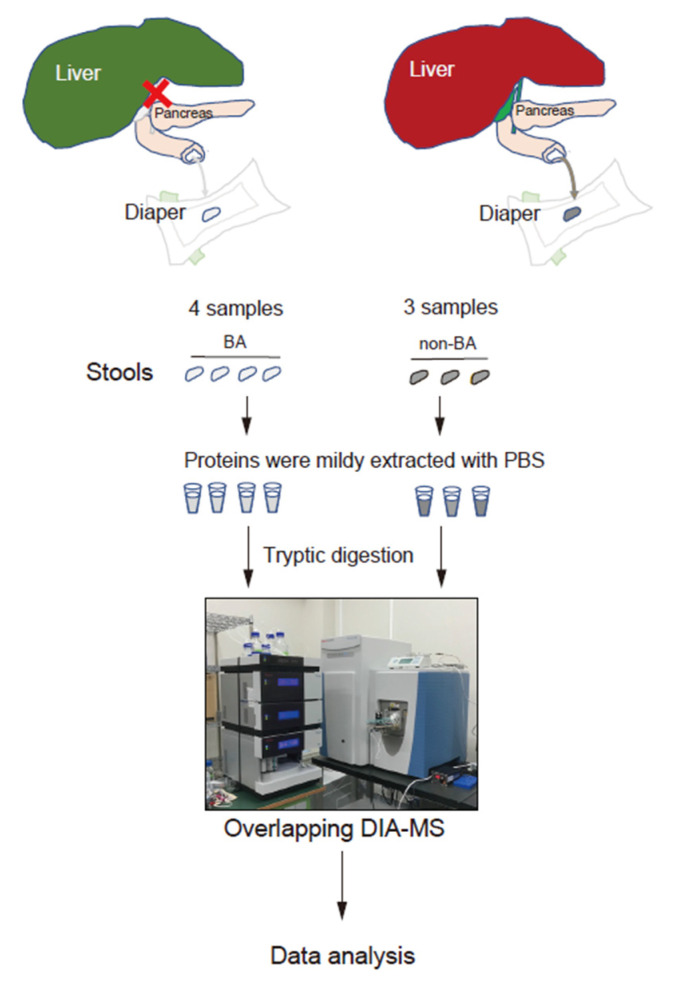
Scheme of stool proteome analysis targeting host-derived proteins. Abbreviations: BA, biliary atresia.

**Figure 2 proteomes-08-00036-f002:**
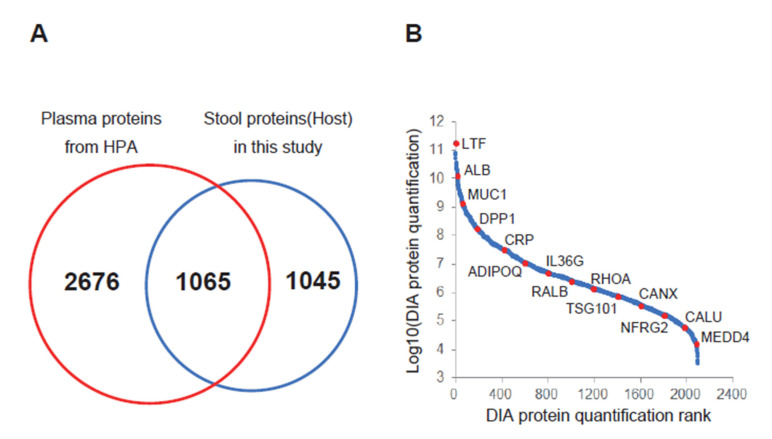
Proteome analysis of stool samples by overlapping data-independent acquisition mass spectrometry (DIA–MS). (**A**) Venn diagram of the number of stool proteins identified from our result and the number of plasma proteins registered in the Human Protein Atlas (HPA; https://www.proteinatlas.org/). (**B**) Dynamic range of our stool proteome analysis. The average protein quantitative values of seven samples were plotted. Representative proteins are indicated by red dots. Abbreviations: ALB, serum albumin; ADIPOQ, adiponectin; CALU, calumenin; CANX, calnexin; CRP, C-reactive protein; DPP7, dipeptidyl peptidase 2; IL36G, interleukin-36 gamma; LTF, lactotransferrin; MUC1, mucin-1; NEDD4, E3 ubiquitin-protein ligase NEDD4; NDRG2, protein NDRG2; RALB, Ras-related protein Ral-B; RHOA, transforming protein RhoA; TSG101, tumor susceptibility gene 101 protein.

**Figure 3 proteomes-08-00036-f003:**
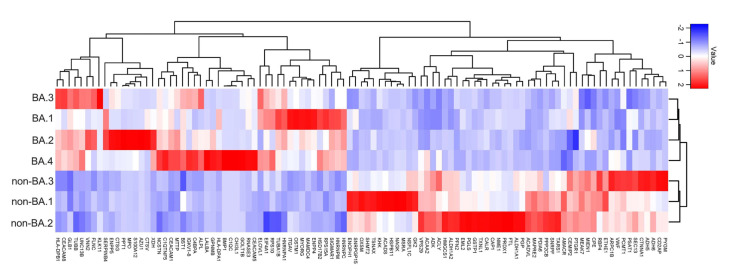
Heatmap showing the relative abundance of proteins in stools between patients with BA versus non-BA patients. Each column represents a single patient. Abbreviations: BA, biliary atresia.

**Figure 4 proteomes-08-00036-f004:**
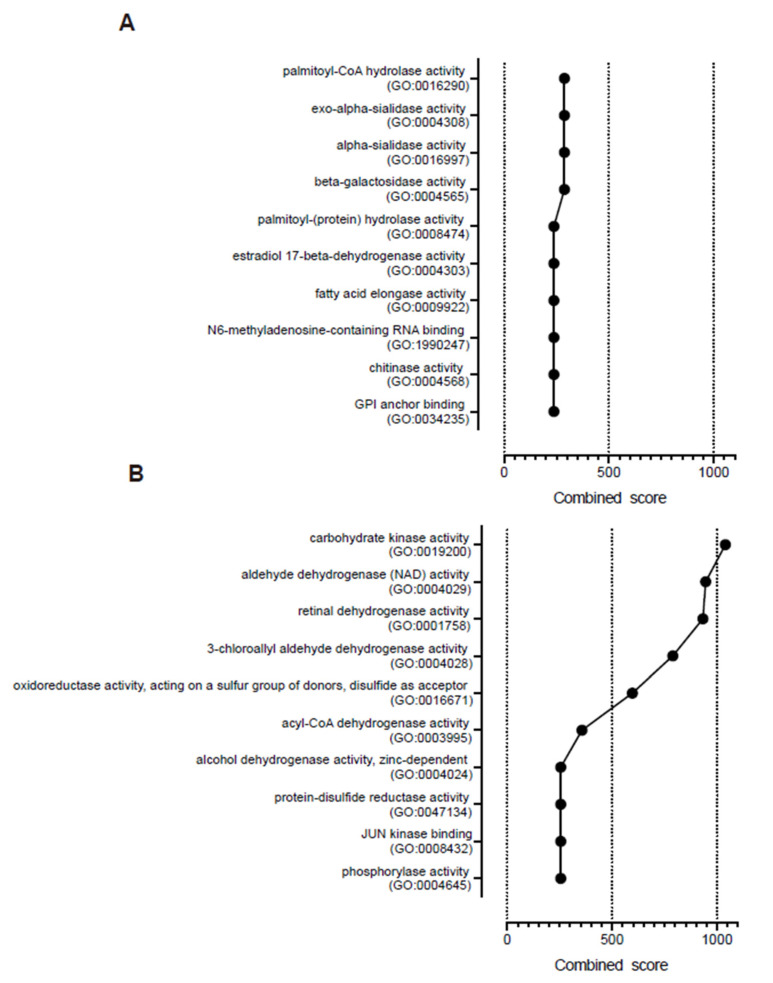
Molecular functions determined by the gene ontology (GO) enrichment analysis. The GO enrichment was analyzed using Enrichr. (**A**) The top ten combined scores of molecular function in the BA-dominant stools. (**B**) The top ten combined scores of molecular function in the non-BA-dominant stools.

**Table 1 proteomes-08-00036-t001:** Clinical features of subjects in this study.

Disease	Sample	Type of BA	Sex		Age	AST	ALT	T-Bil	D-Bil	GGTP
**BA**				RR		13–30	1.0–42	0.4–1.5	0–0.3	13–64
	BA.1	III-c1-ν	M		65	261	238	13	8.9	813
	BA.2	III-a1-ν	M		69	102	80	7.7	5.4	536
	BA.3	III-b1-ν	M		63	186	120	9.1	6.0	276
	BA.4	III-b1-ν	F		25	99	50	4.1	2.8	195
				Ave	56	162	122	8.5	5.8	455
**Disease**	**Sample**	**Type of non-BA**	**Sex**		**Age**	**AST**	**ALT**	**T-Bil**	**D-Bil**	**GGTP**
**non-BA**				RR		13–30	1.0–42	0.4–1.5	0–0.3	13–64
	non-BA.1	NICCD	M		90	241	48	9.4	5.7	172
	non-BA.2	GS	F		38	143	110	5.4	3.2	194
	non-BA.3	VOD	F		112	116	150	25.7	17.0	37
				Ave	80	167	103	14	8.6	134

Abbreviations: Ave, avellage; BA, biliary atresia; NICCD, intrahepatic cholestasis caused by citrin deficiency; GS, gastroschisis; VOD, veno-occlusive disease; AST, aspartate amino transferase (U/L); ALT, alanine aminotransferase (U/L); T-Bil, total bilirubin (mg/dL); D-Bil, direct bilirubin (mg/dL); GGTP, gamma-glutamyl transpeptidase (U/L); RR, reference range. Non-BA: cholestasis other than BA. Age: days after birth.

**Table 2 proteomes-08-00036-t002:** Summary of proteins that are significantly high in stools of BA patients.

Uniprot ID	Protein Name	Gene Symbol	BA/Non-BA
P13497	Bone morphogenetic protein 1	BMP1	∞
P36222	Chitinase-3-like protein 1	CHI3L1	347.3
Q9BW60	Elongation of very long chain fatty acids protein 1	ELOVL1	142.3
P80511	Protein S100-A12	S100A12	72.5
Q14956	Transmembrane glycoprotein NMB	GPNMB	64.4
P50897	Palmitoyl-protein thioesterase 1	PPT1	37.9
P02747	Complement C1q subcomponent subunit C	C1QC	32.4
Q9Y3E0	Vesicle transport protein GOT1B	GOLT1B	25.3
P20160	Azurocidin	AZU1	19.6
P47989	Xanthine dehydrogenase/oxidase	XDH	18.3
O60911	Cathepsin L2	CTSV	17.8
P00709	Alpha-lactalbumin	LALBA	17.0
P20036	HLA class II histocompatibility antigen, DP alpha 1 chain	HLA-DPA1	14.5
P05164	Myeloperoxidase	MPO	11.4
P12724	Eosinophil cationic protein	RNASE3	10.5
P04003	C4b-binding protein alpha chain	C4BPA	9.2
Q9UBX7	Kallikrein-11	KLK11	8.6
P55157	Microsomal triglyceride transfer protein large subunit	MTTP	7.9
P37059	Estradiol 17-beta-dehydrogenase 2	HSD17B2	7.9
P08311	Cathepsin G	CTSG	7.7
Q9UJA9	Ectonucleotide pyrophosphatase/phosphodiesterase family member 5	ENPP5	7.5
P48594	Serpin B4	SERPINB4	6.8
P04440	HLA class II histocompatibility antigen, DP beta 1 chain	HLA-DPB1	6.3
Q6UXC1	Apical endosomal glycoprotein	MAMDC4	5.6
P31997	Carcinoembryonic antigen-related cell adhesion molecule 8	CEACAM8	5.6
O14795	Protein unc-13 homolog B	UNC13B	5.4
A0A0C4DH67	Immunoglobulin kappa variable 1–8	IGKV1–8	5.4
Q6NSJ0	Myogenesis-regulating glycosidase	MYORG	4.9
P16278	Beta-galactosidase	GLB1	4.9
P06731	Carcinoembryonic antigen-related cell adhesion molecule 5	CEACAM5	4.7
Q86WC4	Osteopetrosis-associated transmembrane protein 1	OSTM1	4.7
P13688	Carcinoembryonic antigen-related cell adhesion molecule 1	CEACAM1	4.6
P05186	Alkaline phosphatase, tissue-nonspecific isozyme	ALPL	4.3
O95498	Vascular non-inflammatory molecule 2	VNN2	4.0
P62244	40S ribosomal protein S15a	RPS15A	4.0
Q9Y6 × 5	Bis(5′-adenosyl)-triphosphatase ENPP4	ENPP4	3.8
P09651	Heterogeneous nuclear ribonucleoprotein A1	HNRNPA1	3.4
P52272	Heterogeneous nuclear ribonucleoprotein M	HNRNPM	3.2
Q14315	Filamin-C	FLNC	3.0
Q9BXJ0	Complement C1q tumor necrosis factor-related protein 5	C1QTNF5	2.9
Q92542	Nicastrin	NCSTN	2.8
P06756	Integrin alpha-V	ITGAV	2.6
Q10588	ADP-ribosyl cyclase/cyclic ADP-ribose hydrolase 2	BST1	2.2
Q99720	Sigma non-opioid intracellular receptor 1	SIGMAR1	2.1
P07910	Heterogeneous nuclear ribonucleoproteins C1/C2	HNRNPC	2.1
P46783	40S ribosomal protein S10	RPS10	1.9
P60842	Eukaryotic initiation factor 4A-I	EIF4A1	1.8
P07437	Tubulin beta chain	TUBB	1.8
P68363	Tubulin alpha-1B chain	TUBA1B	1.7

Abbreviations: BA, biliary atresia.

**Table 3 proteomes-08-00036-t003:** Summary of proteins that are significantly high in stools of non-BA patients.

Uniprot ID	Protein Name	Gene Symbol	Non-BA/BA
P11217	Glycogen phosphorylase, muscle form	PYGM	∞
Q15555	Microtubule-associated protein RP/EB family member 2	MAPRE2	∞
Q9Y547	Intraflagellar transport protein 25 homolog	HSPB11	48.6
P02792	Ferritin light chain	FTL	26.7
P02753	Retinol-binding protein 4	RBP4	11.8
O43396	Thioredoxin-like protein 1	TXNL1	11.1
O95336	6-phosphogluconolactonase	PGLS	10.6
P34897	Serine hydroxymethyltransferase, mitochondrial	SHMT2	10.3
Q9UNZ2	NSFL1 cofactor p47	NSFL1C	10.1
Q01581	Hydroxymethylglutaryl-CoA synthase, cytoplasmic	HMGCS1	9.7
Q9NWU1	3-oxoacyl-(acyl-carrier-protein) synthase, mitochondrial	OXSM	9.5
P28332	Alcohol dehydrogenase 6	ADH6	9.5
Q9UJ68	Mitochondrial peptide methionine sulfoxide reductase	MSRA	9.1
P62306	Small nuclear ribonucleoprotein F	SNRPF	8.9
P00352	Retinal dehydrogenase 1	ALDH1A1	7.5
Q15084	Protein disulfide-isomerase A6	PDIA6	6.8
Q9UJ70	N-acetyl-D-glucosamine kinase	NAGK	6.3
O94788	Retinal dehydrogenase 2	ALDH1A2	6.2
P16219	Short-chain specific acyl-CoA dehydrogenase, mitochondrial	ACADS	6.1
O15143	Actin-related protein 2/3 complex subunit 1B	ARPC1B	6.1
O95571	Persulfide dioxygenase ETHE1, mitochondrial	ETHE1	5.9
P15531	Nucleoside diphosphate kinase A	NME1	5.8
P27797	Calreticulin	CALR	5.7
Q06830	Peroxiredoxin-1	PRDX1	5.5
P49748	Very long-chain specific acyl-CoA dehydrogenase, mitochondrial	ACADVL	5.5
Q99598	Translin-associated protein X	TSNAX	5.3
P55263	Adenosine kinase	ADK	5.3
Q01518	Adenylyl cyclase-associated protein 1	CAP1	5.2
Q9Y5K6	CD2-associated protein	CD2AP	5.1
P50053	Ketohexokinase	KHK	5.0
Q6P9B6	MTOR-associated protein MEAK7	MEAK7	5.0
P11766	Alcohol dehydrogenase class-3	ADH5	4.8
P42765	3-ketoacyl-CoA thiolase, mitochondrial	ACAA2	4.4
A8MWD9	Putative small nuclear ribonucleoprotein G-like protein 15	SNRPGP15	4.3
P62714	Serine/threonine-protein phosphatase 2A catalytic subunit beta isoform	PPP2CB	4.3
P35221	Catenin alpha-1	CTNNA1	4.2
P26639	Threonine--tRNA ligase 1, cytoplasmic	TARS1	3.8
O95834	Echinoderm microtubule-associated protein-like 2	EML2	3.6
A6NDG6	Glycerol-3-phosphate phosphatase	PGP	3.4
P55735	Protein SEC13 homolog	SEC13	3.3
Q14410	Glycerol kinase 2	GK2	3.3
P09211	Glutathione S-transferase P	GSTP1	3.3
Q9UBQ0	Vacuolar protein sorting-associated protein 29	VPS29	3.2
P35080	Profilin-2	PFN2	3.1
P53396	ATP-citrate synthase	ACLY	2.9
P16870	Carboxypeptidase E	CPE	2.7
Q9Y617	Phosphoserine aminotransferase	PSAT1	2.5
Q9UHN6	Cell surface hyaluronidase	CEMIP2	2.5
Q9BRT3	Migration and invasion enhancer 1	MIEN1	2.4
Q9UHK6	Alpha-methylacyl-CoA racemase	AMACR	2.3
P22061	Protein-L-isoaspartate(D-aspartate) O-methyltransferase	PCMT1	2.3
P04275	von Willebrand factor	VWF	2.2
Q9UHY7	Enolase-phosphatase E1	ENOPH1	2.0
Q14914	Prostaglandin reductase 1	PTGR1	1.7

Abbreviations: BA, biliary atresia.

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
