# Peer review of "Discovery of Candidate Stool Biomarker Proteins for Biliary Atresia Using Proteome Analysis by Data-Independent Acquisition Mass Spectrometry"

_proteomes, 2020, doi:10.3390/proteomes8040036_

Round 1
Reviewer 1 Report
Agree to accept.
Reviewer 2 Report
The current revision and author’s response to previous comments are acceptable.
This manuscript is a resubmission of an earlier submission. The following is a list of the peer review reports and author responses from that submission.
Round 1
Reviewer 1 Report
This paper describes the proteomic analysis of host-derived proteins in the stool of patients with Biliary atresia and the identification of candidate biomarker proteins. Although the paper is titled "Discovery of stool biomarkers" the validation of the protein as a biomarker may be concluded from the results of future prospective studies or from the results of large-scale analyses. In particular, it may seem premature to conclude from the analysis of protein expression status from three patients alone. I would encourage the authors to reconsider the title.
I am suggesting that the title should be changed following "I would encourage the authors to reconsider the title." Here are some improvements that could be made, including this part
(1) How about changing the title from Identifying Biomarkers to Biomarker Candidate Proteins?
(2) There have been a number of papers on fecal biomarker discovery, so why don't you add the current status of reports on BA and its relationship to this paper as an introduction?
(3) In the method part, there is a description to comparison with "liver tissue", whereas in Figure 2A, the reference is to comparison with "plasma". Please reconcile the description of the method with the results.
Reviewer 2 Report
This review paper by Watanabe et al. demonstrates host-derived stool proteins and developed a DIA-based window spanning proteome analysis using non-invasive stool samples for the discovery of new stool biomarkers for Biliary atresia (BA). The results showed that the profound proteome analysis of stools has a strong potential in the identification of the pathophysiologies and other pediatric diseases, especially in pediatric gastroenterology, as well as the detection of new stool biomarkers for BA. The paper is well written and would serve as important reading to the journal’s readers. The paper has the potential to influence thinking in the field. The authors provided sufficient methodological detail for this review, and the claims were appropriately discussed in the context of the previous literature. A minor suggestion: It would be worth expanding the clinical relevance of the study.
Reviewer 3 Report
The authors try to find potential biomarkers for BA using deep proteome analysis by DIA-MS. The research hypothesis has a certain degree of innovation, but also has obvious defects. The sample size is too small, and the conclusion may be unreliable. Generally, we need to find potential differentially expressed proteins through small samples, and more samples are needed to verify.
In addition, why the study put non-BA group and healthy group together as control group? Generally, we suggest to searched for proteins that were differentially expressed both in BA versus NC and BA versus non-BA firstly. Next, we could remove the proteins that were also differentially expressed in the comparison between the two control groups (non-BA versus NC).